# Data on Herbivore Performance and Plant Herbivore Damage Identify the Same Plant Traits as the Key Drivers of Plant–Herbivore Interaction

**DOI:** 10.3390/insects11120865

**Published:** 2020-12-04

**Authors:** Zuzana Münzbergová, Jiří Skuhrovec

**Affiliations:** 1Department of Botany, Faculty of Science, Charles University, 128 01 Prague, Czech Republic; zuzmun@natur.cuni.cz; 2Institute of Botany, Czech Academy of Sciences, 252 43 Průhonice, Czech Republic; 3Crop Research Institute, 161 06 Prague, Czech Republic

**Keywords:** cost of defense, folivory, Lepidoptera, trait evolution, selection pressure, thistle, knapweed, trichome

## Abstract

**Simple Summary:**

Our aim was to determine the effects of a free-living leaf-chewing generalist caterpillar on leaf damage of several different closely related plant species. Herbivore performance and leaf damage were affected by similar plant traits. Traits related to higher caterpillar mortality (higher leaf dissection, number, length and toughness of spines and lower trichome density) also led to higher leaf damage. Both types of data may be used to identify the key traits driving the interactions. On the other hand, we confirmed that it is very important to carefully distinguish whether the data on leaf damage were obtained in the field or in controlled feeding experiments, as the patterns expected in the two environments go in opposite directions.

**Abstract:**

Data on plant herbivore damage as well as on herbivore performance have been previously used to identify key plant traits driving plant–herbivore interactions. The extent to which the two approaches lead to similar conclusions remains to be explored. We determined the effect of a free-living leaf-chewing generalist caterpillar, *Spodoptera littoralis* (Lepidoptera: Noctuidae), on leaf damage of 24 closely related plant species from the Carduoideae subfamily and the effect of these plant species on caterpillar growth. We used a wide range of physical defense leaf traits and leaf nutrient contents as the plant traits. Herbivore performance and leaf damage were affected by similar plant traits. Traits related to higher caterpillar mortality (higher leaf dissection, number, length and toughness of spines and lower trichome density) also led to higher leaf damage. This fits with the fact that each caterpillar was feeding on a single plant and, thus, had to consume more biomass of the less suitable plants to obtain the same amount of nutrients. The key plant traits driving plant–herbivore interactions identified based on data on herbivore performance largely corresponded to the traits identified as important based on data on leaf damage. This suggests that both types of data may be used to identify the key plant traits determining plant–herbivore interactions. It is, however, important to carefully distinguish whether the data on leaf damage were obtained in the field or in a controlled feeding experiment, as the patterns expected in the two environments may go in opposite directions.

## 1. Introduction

Herbivory is one of the most important factors affecting plant fitness and, thus, represents one of the key selection pressures affecting the evolution of a wide range of plant traits (e.g., [1,2,3,4]). As herbivory is a relationship between two different organisms, its proper understanding requires studying both the plants and the herbivores.

Many previous studies dealt with the effects of plant traits on the performance of herbivores (e.g., [5,6,7,8,9,10,11]) or on plant damage by herbivores [12,13,14]. While both of these approaches have been used to identify plant traits underlying plant defense against herbivores, the conclusions drawn by one or the other approach are expected to differ due to focusing on different aspects of the interaction [4]. However, these two types of responses have less frequently been combined within a single study and our knowledge on their congruence is very limited (but see, e.g., [15]). Information on the correspondence of plant traits predicting plant herbivore damage with the plant traits predicting herbivore performance is, thus, still largely missing [4].

Plant traits related to plant defense against herbivores are known to not only be a result of selection by the herbivores but also reflect the evolutionary history of the plant groups (e.g., [16,17]). Understanding the importance of plant traits for plant–herbivore interactions should, thus, involve exploration of the effects of phylogenetic relationships among the plant species involved. A useful system for such a study is to use sets of closely related species, allowing to minimize the effects of phylogeny [18,19,20,21,22,23]. Closely related plant species sharing a wide range of traits give us the possibility to explore the effects of the varying traits at the background of similar growth form, ecology and evolutionary history and also to reduce the danger of confounding effects of other unstudied plant traits, such as major differences in plant chemistry.

The aim of this study was to determine the effects of a free-living leaf-chewing generalist caterpillar, *Spodoptera littoralis* (Boisduval, 1833), on leaf damage of twenty-four different closely related plant species from the Carduoideae subfamily of Asteraceae. We selected this plant group as it is a highly diverse group of species with similar growth forms and ecology, known for a variety of interactions with herbivores [24]. At the same time, thanks to our previous studies of the system (e.g., [24,25]), we already had a good knowledge of a range of traits of the species. In addition to observing the leaf damage, we also explored the effects of these plant species on growth of the caterpillars. Many previous studies showed that plants of this group host a wide range of herbivores (e.g., [26,27,28,29]) and some of the herbivores have strong documented effects on plant population dynamics (e.g., [27,30,31]). The plants from this group thus possess a wide range of traits enabling them to cope with the herbivore attacks. These traits are related to leaf shape and structure, production of spines and trichomes and to content of nutrients in leaf tissue [24,25]. While the used generalist caterpillar *S. littoralis* may not be fully representative for all possible herbivores feeding on the species, it is, as a representative species, suitable to study plant resistance against generalist herbivores [15,32]. In our previous study [25], we demonstrated that the herbivore preferences of *S. littoralis* largely corresponded to the results-based preferences of a common invasive herbivore, slug *Arion lusitanicus*, in the study area.

By combining the data on plant damage and herbivore response to the plants, we aimed at detecting the correspondence between the two possible measures of response to the plant traits and used this to obtain insights into possible trait selection within the group. It has been repeatedly suggested that contents of secondary metabolites are the key defense mechanisms against herbivores (e.g., [33,34]). However, Carmona et al. [35] suggest that morphological and physical resistance traits are, in fact, more important than secondary metabolites in many systems (e.g., higher trichome density and leaf toughness negatively affected the survival and the development of immature stages). As we assume this is likely true for the plant species studied, we focus on the physical defenses and contents of basic nutrients in plant tissue in this study. We do, however, acknowledge that future explorations of the chemical defenses in the group may strongly extend our knowledge on the overall defense syndromes of the species.

Specifically, we asked the following questions: (i) What is the effect of plant species with different traits on the performance of *S. littoralis*? (ii) What is the effect of *S. littoralis* on leaf damage of the plant species with different traits? (iii) Do the plant traits explaining the plant damage by the herbivores correspond to the plant traits explaining the response of the herbivores to the plants? (iv) Are the conclusions affected by species phylogeny?

We hypothesized that the performance of *S. littoralis* will be affected primarily by nutrient contents in the leaves because chewers prefer higher plant nutritional value. We also assumed that herbivore performance will depend on leaf toughness and leaf hairiness, reflecting the possibility to efficiently chew the leaves and possibly also move across the leaves. We also hypothesized that the traits affecting caterpillar performance will, at the same time, affect leaf damage. Finally, we hypothesized that the effects of plant traits will become weaker after accounting for species phylogeny as some of the traits will be phylogenetically constrained.

## 2. Materials and Methods

### 2.1. Study Plant Species and Study Localities

For the study, we selected all the species of the Carduoideae subfamily of the Asteraceae family occurring in the Czech Republic, Europe, for which we were able to collect enough seeds from 3 populations and successfully cultivate them. This resulted in a set of 24 plant species (*Arctium lappa*, *A. nemorosum*, *A. tomentosum*, *Carduus acanthoides*, *C. crispus*, *C. nutans*, *C. personata*, *Carlina vulgaris*, *Centaurea jacea*, *Cen. macroptilon* subsp. *oxylepis*, *Cen. maculosa*, *Cen. phrygia* 2×, *Cen. phrygia* 4×, *Cen. phrygia* subsp. *pseudophrygia*, *Cen. scabiosa*, *Cirsium acaule*, *Cir. arvense*, *Cir. canum*, *Cir. helenioides*, *Cir. oleraceum*, *Cir. palustre*, *Cir. pannonicum*, *Cir. rivulare* and *Serratula tinctoria*). Nomenclature of the species is unified according to Flora Europaea (http://ww2.bgbm.org/EuroPlusMed/query.asp). Here, 2× and 4× indicate use of diploid and tetraploid cytotypes of the given species, respectively. Each cytotype is treated as a separate species as we have previously shown that the type and intensity of plant–herbivore interaction differ between these two cytotypes [36,37].

For each plant species, we selected three populations within the Czech Republic being at least 20 km apart from each other, resulting in 72 studied populations in total [24]. In each population, we collected a bulk of seeds in 2010. We grew seedlings from these seeds in a temperature-controlled greenhouse (maintaining a temperature above 10 °C) from February to May 2011, with no supplemental light. We selected 5 seedlings from each population and planted them into circular pots (19 cm in diameter, 20 cm deep) in an outside experimental garden, with one seedling per pot, at the end of May 2011. This resulted in a set of 15 pots per plant species, i.e., 360 pots in total. The pots were filled with a mixture of common garden soil and sand at a 1:1 ratio. The plants were kept in the outside experimental garden and regularly watered. By keeping all the plants in exactly the same conditions, we ensured that among-species differences would not be due to differences in the environment of the plants [25]. The plants stayed in the outside experimental garden until initiation of the experiment in July 2012 (see below), so they also experienced natural winter conditions.

### 2.2. Herbivore Used for the Experiment

The herbivore used in the study, the Egyptian cotton leafworm *Spodoptera littoralis* (Boisduval) (Lepidoptera: Noctuidae), feeds on plants of at least 40 plant families, including the Asteraceae family [38]. The extreme polyphagy of *S. littoralis* makes it an excellent bioassay species for comparing leaf palatability across a wide range of plant species [15]. It has also previously been successfully used in a feeding experiment in the same plant subfamily [25]. The results of [25] also indicated that herbivore preferences detected using *S. littoralis* were largely similar to preferences detected using a common invasive herbivore in the study area feeding on species from the group, slug *A. lusitanicus*. Because *S. littoralis* comes from (sub)tropical Africa, it is unlikely to share a coevolutionary history with any of our study’s plant species. Its feeding on the studied plant species should thus reflect the general palatability of the plants [15].

The caterpillars for our experiment originated from a laboratory stock (Laboratory of quarantine organisms, Department of Entomology, Crop Research Institute, Prague, CZ) bred on artificial Stonefly Heliothis diet (Wards Natural Science Inc, Rochester, NY, USA) and were not, thus, adapted to any of our study species. The caterpillars used for the experiment were all of identical age (born within the same day) and were 22 days old when the experiment was initiated. *Spodoptera littoralis* caterpillars of this age have been previously shown to be of suitable size for performing feeding experiments [25,39]. The breeding was carried out in the laboratory at 21.5 °C with fluctuations of 18–25 °C, relative humidity 40–60% and 8 h of darkness per day.

### 2.3. Experimental Design

At the beginning of July 2012, i.e., 1 year after planting the plants into the experimental garden, when most of the species were flowering, we placed one caterpillar of *S. littoralis* on each plant, i.e., 15 individuals per plant species, 5 individuals per population, i.e., 360 plants in total. We performed the experiment using the adult plants as, at this stage, they were large enough to support the caterpillars. In addition, all the plant species grew simultaneously in the second year, and the results were, thus, unaffected by large differences in germination time and growth in the first year of plant growth. Each caterpillar was individually weighed before being placed on the experimental plant. Each plant was then enclosed in a mesh translucent cage supported by a metal frame and attached to the sides of the pot by an elastic band (design previously successfully used in Macel et al. [40]). In this way, we ensured that the caterpillar was unable to leave the plant, but the plant had enough light and space to grow.

The experimental plants were measured prior to placing the caterpillar. Specifically, we estimated the number of leaves, the length of the longest leaf, the number of rosettes and the number of flowering stalks and leaf damage. The experiment ran for 40 days. It was terminated at a stage when the first caterpillars started to pupate. The experiment was terminated simultaneously for all the plants, irrespective of the developmental stage of the caterpillar. While this design meant that we would not know the developmental times of all the caterpillars, it allowed us to compare leaf damage over a constant period of time among all the plant individuals. At the end of the experiment, plant size and damage were determined. It was performed in the same way as at the beginning of the experiment, as described above. Afterwards, *S. littoralis* individuals were weighed to determine their fresh weight, dried to a constant weight and weighed again. As the dry *S. littoralis* weight did not provide any additional insights in addition to its fresh weight, the dry weight is not considered further. We also determined the developmental stage of *S. littoralis* (caterpillar, pupa) at the end of the experiment.

### 2.4. Plant Traits

We used data on a wide range of plant traits obtained in our previous study [24] to explain the performance of the herbivores as well as plant herbivore damage. The study by Münzbergová and Skuhrovec [24] provides a detailed description of the methodology used to measure all the traits. Therefore, we do not provide this description in the main text as it would be identical to what has been published previously. We do, however, provide information on the trait measurements in Appendix A.

Because *S. littoralis* feeds on leaves, all the measured traits are related to leaf characteristics. As plants of this subfamily are known for a variety of physical defense mechanisms, we concentrated on these traits and on leaf nutrient contents. We did not consider leaf chemical defenses, though we acknowledge that their inclusion into the study could provide some additional insights. Specifically, we estimated specific leaf area (SLA), leaf water content, leaf dissection, leaf toughness, spinosity of the leaves, spine length, spine toughness, trichome density, trichome length and nitrogen, carbon and phosphorus content in the leaf biomass and C:N ratio.

SLA is an important proxy of leaf sclerophylly [41] and is also correlated to leaf quality and plant growth rate [42]. It thus represents an important correlate of plant palatability as well as of plant ability to respond to herbivory (e.g., [24,25]). Leaf water content is expected to increase the insect’s ability to eat the leaves [43]. It may also dilute the content of nutrients and secondary metabolites, making the plants less or more suitable for the herbivores [6]. Leaf dissection is a measure of leaf shape, with leaves with entire blades providing more feeding space for invertebrate herbivores without the need to move (e.g., [44]). Leaf toughness correlates with fiber and lignin content in the leaves [45,46] and is likely to indicate herbivore ability to chew the tissue [47,48,49]. Spines (measured as spinosity of the leaves and spine length and toughness) are primarily expected to serve as a defense against vertebrate herbivores (e.g., [41,50,51,52]) but may also affect movements of invertebrate herbivores [24,53]. Trichomes (measured as trichome density and length) are hair-like appendages developed from epidermic cells [54] and are expected to primarily evolve as an anti-herbivore defense (e.g., [55,56,57]), despite also having other functions in the plants [55]. Contents of nitrogen and phosphorus in the leaves reflect the nutritive quality of the leaves [24,58,59]. The content of carbon, especially relative to nitrogen (C:N ratio), reduces plant quality [60]. All the traits were measured on multiple individuals for each population. Our data indicate that the variation in all these traits is much larger among the species than within species. Averages of all the traits per plant species were used as predictors in the final analyses presented below. We did not distinguish among plant populations as we found no significant among-population differences in the plant traits measured.

### 2.5. Plant Damage

To estimate the degree of leaf damage of each individual plant, we counted the number of undamaged leaves, the number of leaves with damage less than 10%, the number of leaves with damage between 10% and 50% and the number of leaves with damage over 50%, classified based on visual estimates of the degree of leaf damage. We used this information to estimate the proportion of damaged leaves, the proportion of leaves with more than 50% damage and the overall degree of leaf damage. Because of strong correlations between all the measures of leaf damage, we eventually used only the proportion of damaged leaves in the tests.

### 2.6. Data Analyses

#### 2.6.1. Plant Traits

The first step to analyze the data included simplification of our trait matrix. Specifically, we checked pair-wise correlations between all the traits. We used the information on correlations among the traits to reduce the number of traits from 13 to 9. These 9 traits included all the traits showing a low (r < 0.65) correlation to all the other traits in the trait dataset. When selecting among traits, we always preferred traits previously more frequently reported to be related to plant interactions with invertebrate herbivores. The final traits used in the analyses included SLA, water content in fresh leaves, phosphorus content in the leaves, C:N ratio in the leaves, leaf dissection, trichome length and density and spine density and toughness. Among-species variation in the trait values ranged from 71% for P content in the leaves to 780% for trichome length.

In addition to using the single traits, we also performed a principal component analysis (PCA), as implemented in Canoco 5.0 [61], using the 9 preselected traits and used position of species on the first two ordination axes as composite plants traits. We constructed the composite traits to obtain the best simple representation of all the 9 traits tested representing species anti-herbivore defense syndromes (e.g., [62,63]). By comparing the predictive power of these composite plant traits to the single traits, we attempted to see the amount of information possibly lost when considering the composite traits only. On the other hand, showing the results for single traits is useful to aid comparison to other studies. The first PCA axis (explaining 32.9% of the total variation) distinguished species with dense short trichomes on the leaves from species with long tough spines and long trichomes on the leaves. The second axis (explaining 22.4% of the total variation) distinguished plants with leaves with high SLA and P and water content from plants with the opposite traits. We used these two PCA axes as alternative predictors of plant–herbivore interactions (later referred to as composite traits). As the position of species on the second axis never had any significant effect, we only present the effects of the first PCA axis, together with the 9 single traits, in the results.

#### 2.6.2. Correlation Among the Dependent Variables

First, we explored pair-wise correlations between the characteristics describing the performance of *S. littoralis* (% survival, fresh weight of surviving individuals and % of individuals which pupated) and the proportion of damaged leaves. All these tests were performed using each plant species as a replicate and the values entering the tests thus represented mean values across all individuals of the given plant species. The relationships were tested using Spearman’s rank correlation.

#### 2.6.3. Determinants of Survival and Growth of *S. littoralis* and Leaf Damage

Because of the quite high mortality of *S. littoralis* in the experiment (63%), we started by determining factors affecting *S. littoralis* survival. Afterwards, we selected only the surviving *S. littoralis* individuals and determined the factors responsible for *S. littoralis* weight and developmental stage (caterpillar or pupae and adult). We merged pupae and adults due to having only 20 adult individuals in the dataset.

Because the primary level of replicate in our dataset is species, we calculated the average of each of the dependent variables per plant species (i.e., *S. littoralis* survival, *S. littoralis* weight, *S. littoralis* developmental stage and leaf damage) and tested the effect of plant traits on these dependent variables. While we initially also analyzed the complete dataset with each individual as a replicate and accounted for the dataset structure using mixed effect models, these tests did not bring any additional insights into the data and are, thus, not presented. Each species originally came from 3 populations and thus we also considered including information on the population into our models. As this did not provide any additional insights, as we only had 5 replicates per population, these results are also not presented further, and we only work with species means further. In this way, we also ensure that the tests directly correspond to the figures, which would not be easily possible if more complex models were used.

The significance of the plant traits, including the composite trait (species position on the first axis of the trait PCA), was first tested separately for each trait. To do this, we compared models with the single trait to an empty model using the Akaike information criterion (AIC) values and calculated ΔAIC for each trait. Afterwards, all the traits were combined in a single total model and selected using bi-direction step-wise regression analyses (based on AIC criteria). All the tests were done using linear models in R 2.14.1 [64]. We used the single trait analysis combined with the step-wise regression analysis because we had relatively few datapoints (24 species) for many predictors (9 traits plus the composite trait). Using regression models with all the dependent variables simultaneously may lead to unreliable, overparametrized models. By assessing significance using the ΔAIC within the step-wise analyses, we corrected for the number of predictors (as ΔAIC is a procedure penalizing for adding an extra term in the model [65]). Thanks to this, the results based on the step-wise selection could be considered as an alternative of the standard regression model followed by Bonferroni’s correction for multiple testing [66].

#### 2.6.4. Importance of Phylogenetic Relationships

To assess the effect of phylogenetic relationships among the species on the patterns observed, we used data on phylogeny of the plant group developed for the purpose of a previous study [67]. The phylogenetic distance matrix describing the relationships among the plant species was decomposed into its eigenvectors using a principal coordinates analysis (PCoA), as suggested by Diniz et al. [68] and Desdevises et al. [69], using the R-package ‘ape’ [70]. The first eigenvector explained 56% of the variability in the data. It was included as a co-variable in all the above analyses in order to correct for phylogenetic autocorrelation. These results are shown together with the results without this co-variable. We also performed separate tests to assess the effect of this phylogenetic eigenvector on all the plant traits as well as on all the dependent variables. In this way, we explored whether the traits and the dependent variables are phylogenetically constrained.

## 3. Results

### 3.1. Correlation among the Dependent Variables

There was no significant correlation between proportion of surviving *S. littoralis*, mean weight of the surviving *S. littoralis* at the end of the experiment and developmental stage of *S. littoralis* and the different plant species (Table 1). Out of these, the strongest relationship, though not significant, was the positive relationship between *S. littoralis* weight and survival. There was also no significant correlation between proportion of surviving *S. littoralis*, mean weight of the surviving *S. littoralis* at the end of the experiment and developmental stage of *S. littoralis* on one side and proportion of damaged leaves on the other side (Table 1).

### 3.2. Determinants of Survival and Growth of S. littoralis

Four out of the nine tested plant traits plus the composite trait had significant effects on *S. littoralis* survival when tested separately (Table 2). The survival decreased with increasing leaf dissection, number of spines, spine toughness (Figure 1a) and the composite trait and increased with increasing trichome density. Out of these, the composite trait had the highest ΔAIC value, followed by trichome density. When all the traits were included into a single model and the best plant traits were selected using a step-wise selection, only spine toughness (Figure 1a) and the composite trait were retained in the final model (Table 2).

*Spodoptera littoralis* weight was significantly negatively affected by leaf dissection (Figure 1b) and the composite trait (Table 2), with leaf dissection having a higher ΔAIC value. When using step-wise selection, only the leaf dissection (Figure 1b) was retained in the model.

Caterpillars of *S. littoralis* have distinctly shorter development periods on plants with larger leaf dissection, i.e., on species with higher *S. littoralis* mortality. They were also able to pupate during the experiment on plants with lower phosphorus content (Figure 1c). This suggests that *S. littoralis* develops faster on less suitable plants (Table 2). Phosphorus content had a higher ΔAIC value and has been the only predictor retained in the overall model (Figure 1c).

### 3.3. Determinants of Leaf Damage

The proportion of damaged leaves was higher in plants with higher C:N ratio (Figure 1d), number of spines, spine toughness (Figure 1e) and trichome length and the composite trait and was lower in plants with more pubescent leaves. The highest ΔAIC value was in the composite trait followed by the trichome density and spine toughness (Table 2). When combined into a single model, only C:N ratio (Figure 1d) and spine toughness (Figure 1e) were retained in the model (Table 2).

### 3.4. Effects of Phylogeny

Phylogenetic relationships of the species had a significant effect on number of spines, spine toughness, leaf hairiness, trichome length and the composite trait, but not on P content, C:N ratio and leaf dissection (Table 2). Phylogeny had also significant effect on *S. littoralis* survival and the proportion of damaged leaves, but not on *S. littoralis* weight and pupation (Table 2). After including phylogeny into the models exploring the effects of species traits, the results remained largely unchanged (Table 2).

## 4. Discussion

The study showed that survival and growth of *S. littoralis* on different plant species of the Carduoidea subfamily could be explained by among-species differences in plant physical defense traits and nutrient contents in the leaves. Furthermore, the proportion of damaged leaves differed between the plant species and could be linked to the differences in the plant traits. All this shows that the plant traits measured are important determinants of the growth and feeding patterns of *S. littoralis* and may, thus, be under strong selection pressure. The traits supporting high *S. littoralis* survival (lower leaf dissection, number, length and toughness of spines and higher trichome density) were the same traits as those determining lower leaf damage of the plants. In contrast to the conclusions of Erb [4], this indicates that different proxies of plant–herbivore interactions may lead to similar conclusions on the importance of plant traits for species defense against herbivores. While the plant traits, performance of *S. littoralis* and plant damage were significantly affected by species phylogenetic relationships, including phylogenetic information into the models had only little effect on the results. This suggests that despite its statistical significance, phylogenetic information had little biological relevance in the system.

### 4.1. Determinants of Survival and Growth of S. littoralis

We identified several plant traits affecting herbivore performance on the plants. Leaf dissection was the only trait affecting all the three variables describing herbivore performance. Caterpillar survival and weight were lower on species with more dissected leaves (Figure 1b). This is in line with the expectation that entire leaves provide more feeding space without the need to frequently move from place to place, being more favorable for invertebrate herbivores [71,72]. Herbivores may also see dissected leaves as leaves already attacked by other herbivores, stimulating their faster pupation [73].

Faster caterpillar development was also observed on plants with lower phosphorus content (Figure 1c), though this did not translate into lower caterpillar weight in our dataset. This result, together with faster pupation on more dissected leaves, suggests that the caterpillars shortened their developmental time in case of less suitable conditions (e.g., [44]). It is also possible that the caterpillars with slower development suffered higher mortality on the less suitable plants. As a result, only caterpillars with faster development survived on the less suitable plants. In contrast, caterpillars with slower development may also have survived on the more suitable plants, resulting in slower mean development on these plants [74,75].

In contrast to previous studies (e.g., [55,76,77]), caterpillars of *S. littoralis* survived better on leaves with higher trichome density (Table 2). This suggests that the presence of trichomes on the leaves created conditions favorable for the survival of the caterpillars. In line with this, Agrawal [5] found a positive effect of trichome density on aphid growth and suggested that the trichomes may provide a beneficial microenvironment for the aphids. A similar explanation may be used in our system, as the weather was extremely hot during the experiment (www.chmi.cz; average temperature in July 2012 = 19.75 °C; in July 2011 = 17.81 °C) and the trichomes may have provided shelter to the caterpillars while feeding on the leaves, at least in the initial stages of caterpillar development, when they are of similar size as aphids. In addition, the species with leaves with higher trichome density could have, themselves, been better protected from the extreme heat (e.g., [78,79]) and, thus, be more palatable. *Spodoptera* survival was also negatively affected by the number of spines and spine toughness.

Out of all the traits, only C:N ratio, water content, SLA and trichome length did not have any significant effect on herbivore performance. This contrasts with studies indicating the importance of C:N ratio and water content in plant leaves for the herbivores [6,80,81,82]. The absence of the effect of these traits may be because herbivore performance is affected by many different leaf traits with very complex effects, where the effect of one trait may be masked by the effects of the other traits [7,83]. It is also possible that the variation in the C:N ratio and water content in our study system was lower than in the previous studies given that we worked with a limited set of closely related species cultivated under uniform conditions (for C:N ratio, the values vary by 116%, and for water content, by 185%). It is also important to note that C:N ratio and trichome length had significant positive effects on leaf damage, as discussed below.

Overall, we found more significant determinants of *S. littoralis* survival than of its growth and length of caterpillar development, indicating that the ability to survive is the key selection factor in our experimental system and that different processes affect different periods in the life of *S. littoralis* caterpillars. This pattern is likely related to the fact that the mortality had been quite high and the data on pupation and growth are, thus, based on the subset of more vital caterpillars which had survived.

### 4.2. Determinants of Leaf Damage

The traits reducing the performance (mainly survival) of *S. littoralis* were, at the same time, traits correlated with higher leaf damage by *S. littoralis* on the plants where the caterpillars survived and vice versa. This seeming controversy could be explained by leaf palatability, as the caterpillars feeding on more nutritious leaves need to feed less, thus causing lower leaf damage and surviving better (e.g., [74,84]). This is true in our experimental system because the caterpillars were restricted to single plant individuals and could not move to feed on more nutritious plants. The limiting factor for leaf damage in our experiment is the incomplete development of some caterpillars in the last instar. In this phenological stage, caterpillars usually cause the most damage and, therefore, the effect of leaf damage can be biased.

Increased leaf damage on plants with a higher C:N ratio (Figure 1d) is, thus, clearly a function of the lower palatability of these plants (similar to, e.g., [60]). Contrasting results of other studies showing higher leaf damage on plants with higher nitrogen and, thus, lower C:N ratio (e.g., [24,85,86]) likely come from a different design of the study. Herbivores are more likely to select plants with a higher nitrogen content in the case of a field study where they are free to select the food plant. In contrast, when left to feed on a limited sample of plants within an experiment, they need to consume more of the plants with less nitrogen to fulfil their nitrogen demands.

Higher damage was also detected on leaves with a higher number of spines and higher spine toughness (Figure 1e), traits simultaneously having the opposite effects on survival. Negative effects of spines on feeding by herbivores is commonly reported for large vertebrate herbivores (e.g., [50,51,52,87,88]). However, studies exploring the effects of spines on invertebrate herbivores are much rarer [24,53]. Presence of spines is likely a trait primarily developed for plant defense against large vertebrate herbivores as these structures can be easily avoided by much smaller invertebrates. As plants with this type of mechanical defense may, in fact, otherwise produce leaves of higher quality [52], it could be expected that plants with larger and tougher spines may be more attractive for invertebrate herbivores. Such a pattern was confirmed in our previous study dealing with the same plant group and looking at natural herbivore damage in the field [24]. In the current study, we also found increasing plant damage, but lower survival of *S. littoralis*, on plants with more, longer and tougher spines. This contrasts with the expectation that tougher leaves have higher quality [24].

One explanation could be related to the fact that *S. littoralis* is a large, fast-growing generalist species requiring a high quantity of high-quality food [15]. In contrast, natural herbivores are generally smaller and more specialized [24,26,28,29]. The contrasting result obtained here in comparison with the previous field study, thus, suggests that the relationship between plant traits and herbivory is very dependent on the type of herbivore tested. Such different preferences of different herbivores were also reported in a range of other studies (e.g., [21,25,89,90], but see [91]) and suggest that a larger range of carefully selected herbivores should be tested to understand the functions of different defenses of plants against herbivores. The mismatch between the results of this study and those of Münzbergová and Skuhrovec [24] could also be because the presence of spines and trichomes could be theoretically correlated with inducible chemical traits, which may be induced in the field but not in this short-term garden experiment (see [83] for a similar reasoning).

Leaf damage increased on leaves with lower density and increased length of trichomes. The effect of trichome density is in line with its opposite effect on caterpillar survival, as explained above. Above, we speculated that denser trichomes provided shelter to the very young caterpillars in the extreme hot weather of the experiment. Well-protected leaves may contain more water and, possibly, also nutrients (e.g., [78,79]), and thus, a lower quantity of leaf material is needed by the caterpillars. It is also possible that the leaves with more trichomes have a lower chemical defense and are, thus, more palatable [92]. Field studies looking at freely moving herbivores showed an opposite result, i.e., lower leaf damage of leaves with more trichomes (e.g., [24,93]), supporting the expectation that trichomes primarily evolved as an anti-herbivore defense (e.g., [55,56,57,94]). Other studies, however, found the opposite and suggested that trichomes may complicate movement of predators on the plant and, thus, provide better conditions for the invertebrate herbivores [95,96]. No such pattern can be expected in our study as our plants were protected from any possible predator attacks. The effect of trichomes on herbivores can be affected not only by the length and density of the trichome but also by several other aspects, such as the type of trichome (e.g., [55,56]). Some trichomes, due to their morphology and composition, can cause specific digestive problems for some groups, such as caterpillars [57,97,98]. This should be taken into account more in further studies, and the morphology of the trichome should be studied in more detail.

### 4.3. Correspondence between Determinants of Herbivore Performance and Plant Damage

Comparison of predictors of herbivore responses to the plants and plant responses to the herbivores indicated that most of the patterns are consistent. Most importantly, traits negatively correlated to caterpillar survival were, at the same time, positively correlated to plant damage and vice versa. This clearly shows that the less suitable food plants, where the herbivores need to feed more in this closed experimental system, at the same time, lead to increased herbivore mortality, despite still having enough food. The same plant traits are expected to lead to lower herbivore damage in these plants in the case of natural conditions where the herbivores can select their food source, as discussed above [24,25]. This suggest that the results of studies using herbivore performance and plant damage as the proxies may lead to similar conclusions on the key plant traits driving the plant–herbivore interactions.

### 4.4. Effects of Phylogeny

Caterpillar survival and leaf damage, but not caterpillar weight and pupation, were significantly affected by species phylogeny. Furthermore, all the plant traits that significantly correlated with these two variables were also significantly related to phylogeny. This seems to suggest that the detected relationships between caterpillar survival and leaf damage and plant traits may be due to some other unmeasured traits closely related to species phylogenetic relationships, such as content of secondary metabolites. This expectation was, however, not supported by our results as the patterns remained largely unchanged after accounting for species phylogeny. This suggests that the patterns detected are not solely due to species relatedness. Phylogenetic conservatism in species traits related to anti-herbivore defense has been shown previously (e.g., [99,100,101]). In line with our results, these studies also suggested that the patterns of plant–herbivore interactions are not solely driven by this phylogenetic conservatism.

### 4.5. Limitations of the Study

Despite the study providing important findings indicating congruency among traits detected as important for plant–herbivore interactions based on herbivore performance and plant damage, it also has many important limitations. First of all, we only worked with physical defense traits and simple leaf chemistry and did not consider possible chemical defenses. As we also analyzed the effects of each trait separately, our results should not be affected by this. Still, understanding the importance of chemical defenses in the same plant group is another important step in the studies of the group to properly understand the drivers of plant–herbivore interactions in the system.

Second, the conclusions of the study are based on a simplified system with a single caterpillar feeding on a single plant individual without possibility to move. This controlled system allowed us to study the direct relationship between herbivore performance and plant traits but did not allow us to consider the fact that herbivores choose the best plant for their feeding. This is an important fact that needs to be considered when interpreting our results, as already explained above.

Third, by exploring performance of the herbivores covered in cages, we also ignored the fact that they could be attacked by predators in natural systems. While this makes our results less realistic, it makes them easier to interpret and understand. By removing one level of complexity, we were able to detect the direct relationship between plant defenses and their palatability. Further experiments may, now, test how the conclusion might be affected by adding additional trophic levels into the system.

## 5. Conclusions

Performance of *S. littoralis* was strongly affected by a wide range of plant traits, suggesting that plant traits are effective in regulating the interaction between plants and herbivores. Interestingly, the plant traits had differential effects on the survival, growth and development of *S. littoralis*, with survival being the most strongly affected. This suggests that different processes affect different periods in the life of *S. littoralis*. The conclusions on the key plant traits driving the plant–herbivore interaction, derived from observations of the herbivores, were very similar to traits identified as important based on data on leaf damage. This suggests that both types of data may be used to identify the key traits determining plant–herbivore interactions. The results also indicate that plant traits affecting plant–herbivore interactions in this controlled experimental study, where the herbivores were restricted to a single plant individual, may point to an opposite direction than effects detected in field studies observing natural plant damage by freely moving herbivores [24,25]. Future studies exploring the importance of plant traits for plant–herbivore interactions thus need to carefully pay attention to the context under which the associations have been observed.

## Figures and Tables

**Figure 1 insects-11-00865-f001:**
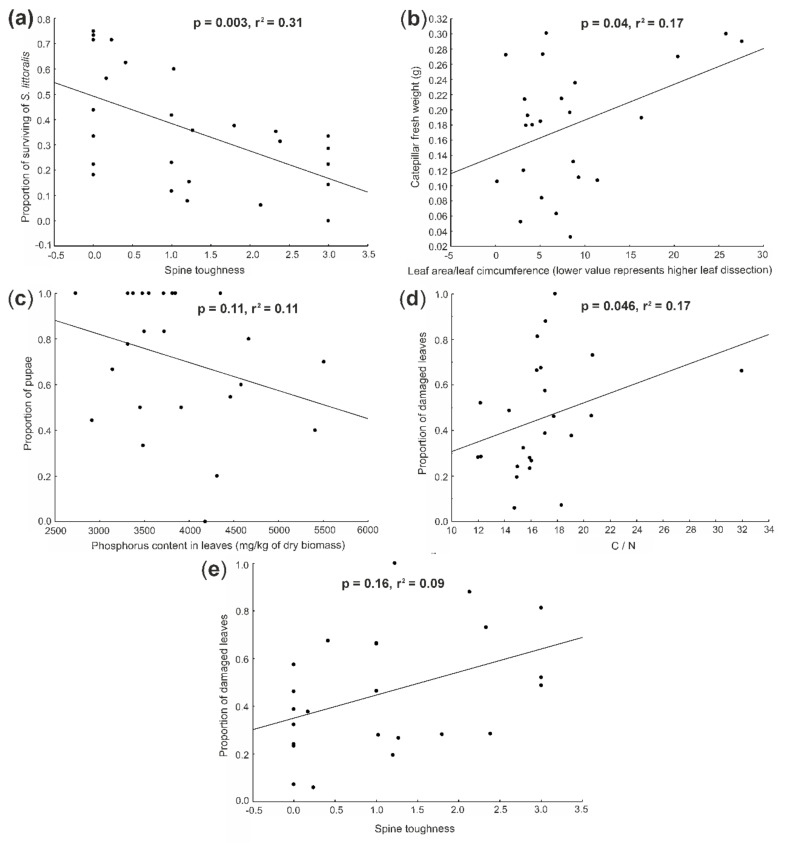
Effect of (**a**) spine toughness on the proportion *S. littoralis* surviving on each tested plant species, (**b**) leaf dissection on *S. littoralis* fresh weight, (**c**) phosphorus content in the leaves on proportion of pupae, (**d**) carbon to nitrogen (C:N) ratio on proportion of damaged leaves and (**e**) spine toughness on proportion of damaged leaves. The figures show all the significant relationships identified in Table 2, which were retained in the final models based on the step-wise selection procedure (marked S in Table 2, except for the composite trait). The datapoints represent means of the predictor and response variables for each species. The significance values are shown in the graphs.

**Table 1 insects-11-00865-t001:** Pair-wise correlation between the dependent variables used in the experiment, assessed using Spearman rank order correlations. None of the relationships are significant (*p* > 0.05). The test was done at the level of single plant species, and thus, N = 24.

	*S. l.* Survival	*S. l.* Weight-Fresh	*S. l.* Pupation	Prop. dam. Leaves
*S. l.* survival	–	0.38	−0.15	−0.33
*S. l.* weight-fresh	0.38	–	0.01	−0.21
*S. l.* pupation	−0.15	0.01	–	0.28
Prop. dam. leaves	−0.33	−0.21	0.28	–

*S.l.* = *Spodoptera littoralis*.

**Table 2 insects-11-00865-t002:** The effect of phylogenetic relationships of the species (Phylogeny) and plant traits includes the composite trait (PCAtrait1) on *S. littoralis* (*S.l.*) survival, fresh weight and pupation and on the proportion of damaged leaves. N = 24 in all cases. The values in the table represent the ΔAIC value from a comparison of an empty model and a model including the given predictor. Negative ΔAIC values indicate that the model improved after adding the given predictor, and a more negative ΔAIC value indicates higher improvement of the model. Negative ΔAIC values are shown in bold. * indicates that the given predictor had a significant (*p* ≤ 0.05) effect when tested separately. S indicates that the predictor was retained in a model selecting predictors using the bi-directional step-wise selection procedure. Underlined values are significant even after accounting for phylogeny. The signs in brackets show the direction of the significant relationships. The line “Phylogeny” shows the effects of phylogeny on the plant traits. The test was performed at the level of single plant species.

	*p*	C:N Ratio	Water Content	SLA	Leaf Dissection	No. Spines	Spine Toughness	Trichome Density	Trichome Length	PCAtrait1	Phylogeny
Phylogeny	0.002	0.006	0.017	0.004	0.012	**−0.144 ***	**−0.283 ***	**−0.022 ***	**−0.066 ***	**−0.219 ***	
*S. l.* survival	0.272	0.819	0.682	0.659	**(−) −1.687 ***	**(−) −1.611***	**(−) −1.590 *S**	**(+) −2.026 ***	0.109	**(−) −4.710 *S**	**−0.730 ***
*S. l.* weight-fresh	0.04	0.038	0.037	0.008	**(−) −0.052 *S**	0.001	0.013	0.025	0.014	**(−) −0.034 ***	0.024
*S. l.* pupation	**(−) −0.013 S**	0.103	0.148	0.141	**(+) −0.009 ***	0.088	0.153	0.149	0.082	0.160	0.158
Prop. dam. leaves	0.365	**(+) −0.174 S**	0.242	0.315	0.145	**(+) −0.188**	**(+) −0.499 *S**	**(−) −0.664 ***	**(+) −0.409 ***	**(+) −0.673 ***	**−0.057 ***

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
