# Peer review of "Data on Herbivore Performance and Plant Herbivore Damage Identify the Same Plant Traits as the Key Drivers of Plant–Herbivore Interaction"

_insects, 2020, doi:10.3390/insects11120865_

Round 1
Reviewer 1 Report
The manuscript entitled “Data on herbivore performance and plant herbivore damage identify the same plant traits as the key drivers of plant-herbivore interaction” describes a study that investigates the effects of different plant traits on herbivore performance and leaf damage. The study is well designed and the authors discuss properly any potential limitations that the experiment could have. One major flaw of the ms is that it is missing Table 2, which is probably the most important part of the results section. In addition, the conclusion that “field and controlled feeding experiment show opposite pattern” should be supported by previous studies.
Comments
Line 24: “and the effect of these plant species on the caterpillar growth”
Lines 24-25: We used a wide range of physical defense leaf traits and leaf nutrient content as plant traits”
Lines 30-32: Please rewrite the sentence to make better sense.
Line 33: Interactions
Lines 58-59: Please combine in one sentence.
Lines 66-71: The authors should report here the traits measured, so that the reader would know the traits at this point and would only open the other publication to read about the methods.
Methods section: Please report the total number of plants used (i.e. how many replicates per treatment).
Line 186: Please replace approximate. Maybe a sentence with “correlate” or “indicate” would be preferable.
Lines 223-228: What is the variance explained by each PCA axis? Please report.
Lines 292: Replace with “significant effects”
Lines 292-: The authors have not included Table 2 (which is very important for the ms)
Lines 322-328: Please add a metric of significance for the plotted lines (e.g. p-values, R2 etc)
Lines 372-375: Is the size of S. litorralis (at the first growth stages) comparable to aphids? Can the trichomes provide microclimatic shelter to the caterpillars?
Lines 375-376: How possible is this explanation, considering that water content was not significant at all?
Lines 383-385: Please rewrite for clarity.
Line 435: Erase “of” in “also of nutrients”.
Lines 443-449: The authors should either report the non-significant effects or delete this part of the experiment.
Lines 454-455: The authors should make clear that even on the heavily damaged plants, there was enough food until the end of the experiment. In other words, the authors should make clear that the cause of mortality was not starvation.
Lines 455-457: This is very important. Please add a reference.
Lines 499-502: Considering that this is one of the major conclusions of this study, the authors should support the second part of it with references (please see the comment above).
Line 524: Is this correct (around the leaf or inside the leaf?)?
Author Response
Dear reviewer,
thank you very much for the very valuable comments that helped us improve our manuscript. We respond to all your comments. These all responses to individual questions are always given under each your comments.
Comments
Line 24: “and the effect of these plant species on the caterpillar growth”
*** Done
Lines 24-25: We used a wide range of physical defense leaf traits and leaf nutrient content as plant traits”
*** Done
Lines 30-32: Please rewrite the sentence to make better sense.
*** Done
Line 33: Interactions
*** Done
Lines 58-59: Please combine in one sentence.
*** Done
Lines 66-71: The authors should report here the traits measured, so that the reader would know the traits at this point and would only open the other publication to read about the methods.
*** The information has been added.
Methods section: Please report the total number of plants used (i.e. how many replicates per treatment).
*** The information has been added.
Line 186: Please replace approximate. Maybe a sentence with “correlate” or “indicate” would be preferable.
*** Done
Lines 223-228: What is the variance explained by each PCA axis? Please report.
*** The information has been added.
Lines 292: Replace with “significant effects”
*** Done
Lines 292-: The authors have not included Table 2 (which is very important for the ms)
*** Corrected, we called it Table 3, but it was Table 2.
Lines 322-328: Please add a metric of significance for the plotted lines (e.g. p-values, R2 etc)
*** The information has been added.
Lines 372-375: Is the size of S. litorralis (at the first growth stages) comparable to aphids? Can the trichomes provide microclimatic shelter to the caterpillars?
*** The information has been added. The first two instars of S. littoralis are of similar size as aphid.
Lines 375-376: How possible is this explanation, considering that water content was not significant at all?
*** This is not contradictory to water content as water content was a species trait – so was measured previously and not within this experiment. So water content reflects how much water do leaves of the species contain under standardized well-watered conditions.
Lines 383-385: Please rewrite for clarity.
*** The sentence has been changed.
Line 435: Erase “of” in “also of nutrients”.
*** Done
Lines 443-449: The authors should either report the non-significant effects or delete this part of the experiment.
*** Delete it
Lines 454-455: The authors should make clear that even on the heavily damaged plants, there was enough food until the end of the experiment. In other words, the authors should make clear that the cause of mortality was not starvation.
*** The information has been added.
Lines 455-457: This is very important. Please add a reference.
*** The information has been added. We get it from the previous experiments (Kuglerová et al. 2019, Münzbergová and Skuhrovec 2013).
Lines 499-502: Considering that this is one of the major conclusions of this study, the authors should support the second part of it with references (please see the comment above).
*** The information has been added. We get it from the previous experiments (Kuglerová et al. 2019, Münzbergová and Skuhrovec 2013).
Line 524: Is this correct (around the leaf or inside the leaf?)?
*** Around the leaf
Reviewer 2 Report
The manuscript “Data on herbivore performance and plant herbivore damage identify the same plant traits as the key drivers of plant-herbivore interaction” by Münzbergová and Skuhrovec explores the effects of variation in plant traits (nutrient chemistry and physical defenses) on plant damage by, and performance of, the Egyptian cotton leafworm. The manuscript is well written and I found the narrative quite clear, with only minor issues with respect to grammar and conciseness. The authors have also measured an extensive array of variables, which lends itself quite nicely to a comprehensive analysis of trends and relationships. Unfortunately, major issues with the experimental approach limit the value and interpretation of the data collected. Below I have included both general comments highlighting significant drawbacks of the study design, as well as a series of specific comments to help improve clarity and flow of the text.
General Comments
I’m not convinced that choosing closely-related species was the way to go. More distantly related species would not only provide a wider range of trait values, but would make for a more informative phylogenetic analysis as well as more useful contrast between trait effects and phylogenetic relationships. This should be acknowledged, and the authors will need to add compelling rationale as to why more distantly-related species would have been the weaker/wrong choice for this experiment.
Ending the experiment after pupae first began to appear severely hinders the interpretation of your data. It is understandable that you would want to standardize feeding time to properly determine plant damage, but for a proper assessment of caterpillar performance, this experiment should have included a second assay where caterpillars are simply allowed to feed until they either die or pupate/eclose to adult form. Not only is the assessment of “proportion pupated” incomplete because of this, but your fresh weight measurement spans multiple development stages, hindering the interpretation of this response. Because of these decisions, you are unable to address questions such as “does pupating faster come at the cost of smaller adults?” Also, mortality cannot be properly explored as you are not assessing all individuals at the same stage – how much more mortality would the slowly developing larvae suffer if you allow them the opportunity to develop until death/pupation/eclosion? It is apparent that this “snapshot in time” approach to assess performance has significant drawbacks. Having two separate assays to look at plant damage and herbivore performance individually would have been a better choice, and the addition of the latter is likely necessary to fully assess the herbivore responses.
Specific Comments
L15 Just how variable was dissection for a given species? Was it clear cut among all 24? A picture of the dissection gradient would be quite useful. Also, it would be useful if you discussed some of these relevant traits in the intro. After the abstract, dissection, for instance, isn't mentioned again until the methods
L24 and elsewhere – Various grammar issues to revise e.g. We used “a” wide range….
L34-35 – This isn’t universally true. Should be qualified with a “MAY go in opposite directions”
L48 I disagree with this sentiment of “only rarely” as there are several papers that have looked at both (e.g. Mopper and Simberloff 1995; Jamieson et al. 2015, among others) - the wording should be changed e.g. less frequently
L61 Introduce full taxonomic name here
L69 Revise to fix the flow here
L76 A couple lines should follow this sentence, just outlining some physical defenses that have previously been shown to matter. That way the reader gets a better sense of the rationale behind the traits you chose (i.e. leaf dissection has shown to X, likewise, needles cause Y)
L87-88 This is a circular argument. Is there a better reason/prediction as to WHY nutrient content would matter more than say, physical defenses? water content? Etc?
L92 The herbivore affecting the plant is clearly obvious and is thus superfluous
L123 and elsewhere – manuscript can be made more concise e.g. “is known to feed on” can just be “feeds on”
L250 “easily possible incase” is awkward phrasing
L302 A bit vague, particularly considering you fully spell out the effect direction in the subsequent sentences. This should be removed.
L304-305 "likely to pupate" and "time until pupation" are two different variables, yet here and elsewhere you are conflating them. Likelihood suggests a probability analysis (e.g. binomial regression) which you did not do. This should be rectified throughout the text. You also don’t technically show “time until pupation” or “development time” which of course would have required continuing the experiment for the slower developing individuals
L338 As you discuss individual results, you should be citing figures for reader reference
L344 Why word this statement in such a vague way? The data should provide you more definitive answers
L355 perhaps "on SPECIES with more dissected leaves" would work better here as the current phrasing could suggest that more dissected plants within a species, or even more dissected leaves on a given plant lead to these results
L360 Your results don't speak to development time per se, so you shouldn't be using dev time and % pupation interchangeably
L361 But you don't know how heavy the pupae would be for those that were still larvae when the experiment was ended, so you can't conclude this. This is very much as apples to oranges comparison
L363 what do you mean by "also possible"? As in this wasn't assessed, or too difficult to interpret based on your experimental approach? Either way, mortality can't be properly assessed as not all individuals were provided an opportunity to get to the "pupate or die" stage
L365-367 This statement is unclear
L368 Survival data is ambiguous as individuals are spread across multiple life stages
L373 include the actual statistics here, and how they vary from normal rather than simply referencing a URL. That just leads to the main page of the website, and then requiring the reader to do all the leg work to extract and analyze the weather data
L378-379 superfluous and can be removed
L385-386 This is mentioned above regarding your choice of closely related species. It would definitely be worth discussing what variation there was in "previous studies" to give a sense of how restricted your trait values are
L389-390 You wouldn't know about plant species as a driver of pupation as you didn't allow for "death or pupation" to be reached
L391 but you aren't properly analyzing similar stages across plant species, so you aren't able to infer this
L395 Remove "The traits detected as"
L420-421 Reference?
L421 What are "natural herbivores"? As in not coming from a colony?
L450-459 Would be more useful if this could be put in the context of published literature
L491-504 You simultaneous conclude that the data are valuable in terms of predicting plant damage from herbivore performance responses (and vise versa), but at the same time may not at all represent responses from free-living herbivores, thus making the data invalid. Some revision here to simply stress take-home messages and implications of the observed relationships in your study would probably be best
Figure 1 - All Y-axis titles need to be revised to improve clarity except for (b). Are linear fits the most appropriate for all of these relationships? Axis font is really small and should be revised. Also, please add P-values and R-squared values to each panel. Despite significance being referenced in the AIC table, it would be useful to have some inferential statistics here as opposed to simply eyeballing the relative strength of the various relationships
Table 3 - “and thus N = 24” can be removed from the end as you already mention that earlier in the caption
References – must be revised to be properly standardized e.g. Erb (2018) entry has multiple capitalized words in the title. Same for entry “12”, among others
Author Response
Dear reviewer,
thank you very much for the very valuable comments that helped us improve our manuscript. We respond to all your comments. These all responses to individual questions are always given under each your comments.
General Comments
I’m not convinced that choosing closely-related species was the way to go. More distantly related species would not only provide a wider range of trait values, but would make for a more informative phylogenetic analysis as well as more useful contrast between trait effects and phylogenetic relationships. This should be acknowledged, and the authors will need to add compelling rationale as to why more distantly-related species would have been the weaker/wrong choice for this experiment.
*** We fully agree that if we used more distantly-related species, we will certainly achieve many times greater variability in the studied traits. But that wasn't our only goal, we wanted to find out how it works in a close-related group. By studying closely related species, we were able to study the effect of the specific traits, without confounding effects of other unstudied plant traits, such as major differences in plant chemistry, growth form or ecology. To make this clearer, we extended the explanation of this issue given in the introduction.
Ending the experiment after pupae first began to appear severely hinders the interpretation of your data. It is understandable that you would want to standardize feeding time to properly determine plant damage, but for a proper assessment of caterpillar performance, this experiment should have included a second assay where caterpillars are simply allowed to feed until they either die or pupate/eclose to adult form. Not only is the assessment of “proportion pupated” incomplete because of this, but your fresh weight measurement spans multiple development stages, hindering the interpretation of this response. Because of these decisions, you are unable to address questions such as “does pupating faster come at the cost of smaller adults?” Also, mortality cannot be properly explored as you are not assessing all individuals at the same stage – how much more mortality would the slowly developing larvae suffer if you allow them the opportunity to develop until death/pupation/eclosion? It is apparent that this “snapshot in time” approach to assess performance has significant drawbacks. Having two separate assays to look at plant damage and herbivore performance individually would have been a better choice, and the addition of the latter is likely necessary to fully assess the herbivore responses.
*** Yes, we fully agree that our experimental setup does not allow us to evaluate some questions such as whether the speed of development affected the size of the adult, etc. In order to answer these different questions, we would have to have several concurrent attempts, as you rightly noted. We decided that our aims would not be so ambitious, and the questions were directed more at plant damage, plan traits, not the effect on the overall development of the insect, possibly its further viability, etc. Of course, our experimental setup is not perfect and can show only some information according to this research.
According to your right comments, we will be more precise with the explanation of our data, especially the ability of the pupation versus the length of caterpillar development. We corrected mainly the comments around the pupa. Instead of pupation, we comment here on the speed of caterpillar development. To clarify, we have always explained or commented in detail on individual points in Specific Comments (see below).
Specific Comments
L15 Just how variable was dissection for a given species? Was it clear cut among all 24? A picture of the dissection gradient would be quite useful. Also, it would be useful if you discussed some of these relevant traits in the intro. After the abstract, dissection, for instance, isn't mentioned again until the methods
*** Into the methods, we added the following information. “Our data indicate that the variation in all these traits is much larger among the species than within species.“ This is true not only for the dissection, but also for the other traits studied.
Into the introduction, we added the comments to some plant traits (the idenitcal comments as to L76).
L24 and elsewhere – Various grammar issues to revise e.g. We used “a” wide range….
*** We corrected various grammar issues.
L34-35 – This isn’t universally true. Should be qualified with a “MAY go in opposite directions”
*** The sentence has been corrected it.
L48 I disagree with this sentiment of “only rarely” as there are several papers that have looked at both (e.g. Mopper and Simberloff 1995; Jamieson et al. 2015, among others) - the wording should be changed e.g. less frequently
*** The sentence has been corrected it.
L61 Introduce full taxonomic name here
*** The information has been added.
L69 Revise to fix the flow here
*** The sentence has been corrected it.
L76 A couple lines should follow this sentence, just outlining some physical defenses that have previously been shown to matter. That way the reader gets a better sense of the rationale behind the traits you chose (i.e. leaf dissection has shown to X, likewise, needles cause Y)
*** We have added some characters in the sentence. More details to each character is given in Material and Methods – chapter 2.4. Plant traits (lines 173–204).
L87-88 This is a circular argument. Is there a better reason/prediction as to WHY nutrient content would matter more than say, physical defenses? water content? Etc?
*** The sentence has been changed it.
L92 The herbivore affecting the plant is clearly obvious and is thus superfluous
*** The sentence has been changed it.
L123 and elsewhere – manuscript can be made more concise e.g. “is known to feed on” can just be “feeds on”
*** The sentence has been corrected it. The presence of „is known“ was given due to emphaze of our official knowledges.
L250 “easily possible incase” is awkward phrasing
*** The part of sentence has been changed it.
L302 A bit vague, particularly considering you fully spell out the effect direction in the subsequent sentences. This should be removed.
*** Delete it
L304-305 "likely to pupate" and "time until pupation" are two different variables, yet here and elsewhere you are conflating them. Likelihood suggests a probability analysis (e.g. binomial regression) which you did not do. This should be rectified throughout the text. You also don’t technically show “time until pupation” or “development time” which of course would have required continuing the experiment for the slower developing individuals
*** Yes, we fully agree with these differences between these two terms. We will comment on the pupation more to the length of caterpillar development, which is distinctly highly affected. Our experimental setup is not perfect and can show only some information according to this research. We have the possibility to study only some aims, and we chose this direction. Now, we will be more precise with the explanation of our data, especially ability the pupation versus the length of caterpillar development; and that we know the effect on caterpillars and not commonly on herbivores.
L338 As you discuss individual results, you should be citing figures for reader reference
*** The citations of references have been added.
L344 Why word this statement in such a vague way? The data should provide you more definitive answers
*** The exact traits have been in the brackets.
L355 perhaps "on SPECIES with more dissected leaves" would work better here as the current phrasing could suggest that more dissected plants within a species, or even more dissected leaves on a given plant lead to these results
*** The sentence has been corrected it.
L360 Your results don't speak to development time per se, so you shouldn't be using dev time and % pupation interchangeably
*** The sentence has been changed. We comment only the time of caterpillr development. our results allow to analyze this.
L361 But you don't know how heavy the pupae would be for those that were still larvae when the experiment was ended, so you can't conclude this. This is very much as apples to oranges comparison
*** But we did not comment anywhere on the weight of the pupa, here we comment on the caterpillar size and the speed of caterpillar development (subsequently time to pupation) and thir mortality. Everything was edited from the previous your comments.
L363 what do you mean by "also possible"? As in this wasn't assessed, or too difficult to interpret based on your experimental approach? Either way, mortality can't be properly assessed as not all individuals were provided an opportunity to get to the "pupate or die" stage
*** We mean that it could not be fully evaluated. However, we think that based on other data, we can form this hypothesis.
L365-367 This statement is unclear
*** The sentence has been changed.
L368 Survival data is ambiguous as individuals are spread across multiple life stages
*** Reworded to caterpillar mortality, which corresponds to our data.
L373 include the actual statistics here, and how they vary from normal rather than simply referencing a URL. That just leads to the main page of the website, and then requiring the reader to do all the leg work to extract and analyze the weather data
*** We add the average temperature in July 2012 = 19.75°C; and also in July 2011 = 17.81°C.
L378-379 superfluous and can be removed
*** Removed
L385-386 This is mentioned above regarding your choice of closely related species. It would definitely be worth discussing what variation there was in "previous studies" to give a sense of how restricted your trait values are
*** The range of values is provided here. We also methods range of variation of all the values in the methods.
L389-390 You wouldn't know about plant species as a driver of pupation as you didn't allow for "death or pupation" to be reached
*** Reworded to length of caterpillar development, which corresponds to our data.
L391 but you aren't properly analyzing similar stages across plant species, so you aren't able to infer this
*** Reworded to caterpillar life.
L395 Remove "The traits detected as"
*** Delete it
L420-421 Reference?
*** We add the reference about large study about Spodoptera littoralis.
L421 What are "natural herbivores"? As in not coming from a colony?
*** We mean native hervivores („wild“). This specie does not occur in our region.
L450-459 Would be more useful if this could be put in the context of published literature
*** This part was partly corrected, and we add also some references.
L491-504 You simultaneous conclude that the data are valuable in terms of predicting plant damage from herbivore performance responses (and vise versa), but at the same time may not at all represent responses from free-living herbivores, thus making the data invalid. Some revision here to simply stress take-home messages and implications of the observed relationships in your study would probably be best
*** We do not agree that these results contradict each other. On the contrary, previous studies fully support this. It is clear that in a closed system, the results will never be fully identical to an open system. In a closed system, we are deprived of interactions between species or individuals, and therefore, it can never be the same. Therefore, it is very important to draw attention to the conditions of each experiment and emphasize it.
Figure 1 - All Y-axis titles need to be revised to improve clarity except for (b). Are linear fits the most appropriate for all of these relationships? Axis font is really small and should be revised. Also, please add P-values and R-squared values to each panel. Despite significance being referenced in the AIC table, it would be useful to have some inferential statistics here as opposed to simply eyeballing the relative strength of the various relationships
*** We add the P-values and R for each graph. Axis font size was given as it is normal for the Journal.
Table 3 - “and thus N = 24” can be removed from the end as you already mention that earlier in the caption
*** Delete it
References – must be revised to be properly standardized e.g. Erb (2018) entry has multiple capitalized words in the title. Same for entry “12”, among others
*** The references have been corrected
Reviewer 3 Report
The manuscript by Münzbergová and Skuhrovec deals with an important question in evolutionary ecology on dissecting trait variation in herbivory through host plant functional traits. Their results are interesting, methods are detailed and results have been well discussed in context. While I don't have any major issues with the manuscript, I find there are some concerns that need to be addressed. I thoroughly enjoyed reading the manuscript, and the statistical analyses.
- why did the authors only use the mass of caterpillar rather than relative mass gain? Mass gain is a much more preferred index since it removes any initial mass variation and developmental variation within the caterpillar population. I suggest that the authors either add an additional analysis or explain this.
- I wasn't able to understand the rationale behind using Asteraceae as the family of focus, maybe make it clear.
- The possible reasons for trichomes being ineffective as a defense is interesting. However, this doesn't make much sense without a detailed classification of trichomes and examining their post ingestive roles. So it may be premature to say that they are ineffective. This should be addressed in the discussion. Also, trichome can impale soft-bodied insects. It would have been interesting if the authors would have examined the effect of these trichomes on different instars of this caterpillar.
- Line 22: needs more exploration…might be a better phrase than clarification
- Herbivore details are missing in abstract
- Some of the recent literature on spines and caterpillars, trichomes, and their pre and post ingestive roles, latex, and other surface defenses can be added to improve the manuscript.
- Unable to see the detailed description of different plant traits and their relative intensity in each of the plant species made it difficult to visualize the results. Maybe add that.
Author Response
Dear reviewer,
thank you very much for the very valuable comments that helped us improve our manuscript. We respond to all your comments. These all responses to individual questions are always given under each your comments.
- why did the authors only use the mass of caterpillar rather than relative mass gain? Mass gain is a much more preferred index since it removes any initial mass variation and developmental variation within the caterpillar population. I suggest that the authors either add an additional analysis or explain this.
*** At the beginning of the experiment, the caterpillars were not weighed, but they were of the same age (see Methodology).
- I wasn't able to understand the rationale behind using Asteraceae as the family of focus, maybe make it clear.
*** This information has been added.
- The possible reasons for trichomes being ineffective as a defense is interesting. However, this doesn't make much sense without a detailed classification of trichomes and examining their post ingestive roles. So it may be premature to say that they are ineffective. This should be addressed in the discussion. Also, trichome can impale soft-bodied insects. It would have been interesting if the authors would have examined the effect of these trichomes on different instars of this caterpillar.
*** We add some more comments about trichomes and we add also some literature (see lines 448-453).
- Line 22: needs more exploration…might be a better phrase than clarification
*** Done
- Herbivore details are missing in abstract
*** This information has been added.
- Some of the recent literature on spines and caterpillars, trichomes, and their pre and post ingestive roles, latex, and other surface defenses can be added to improve the manuscript.
*** We add also some recent literature (see lines 448-453).
Andama, J.B.; Mujiono, K.; Hojo, Y.; Shinya, T.; Galis, I. Non-glandular silicified trichomes are essential for rice defense against chewing herbivores. Plant Cell Environ. 2020, pce.13775.
Hall, C.R.; Dagg, V.; Waterman, J.M.; Johnson, S.N. Silicon Alters Leaf Surface Morphology and Suppresses Insect Herbivory in a Model Grass Species. Plants 2020, 9, 643.
- Unable to see the detailed description of different plant traits and their relative intensity in each of the plant species made it difficult to visualize the results. Maybe add that.
*** This is fully stated in Appendix A (after the Conclusion). We think this long text would unnecessarily disturb in the main text. If the reviewer and editor insist, we can move part of the text into the methodology.
Round 2
Reviewer 2 Report
The authors have done a commendable job of improving several aspects of the manuscript. However, I remain unconvinced that the major issues have been adequately addressed. As mentioned, ending the experiment at the first sign of pupation severely hindered interpretation of the insect performance data as you do not have proper standardization for many of your response variables. For instance, you indicate that caterpillar development time varied by plant species. Accordingly, when you took fresh weight measurements, individuals from some plant species would be more advanced in their development than individuals from other plant species. In other words, you would end of with a phenology spectrum with different plant species containing different compositions of early-, mid-, and late-instar individuals. Because your measurement does not differentiate between these, the response conflates phenological variation in weight with stage-specific insect responses to plant quality. It would be much more appropriate to compare L6 weights, for example, or pupa weights, etc. You are also not consistent with how you refer to your response variables. In your figure and your response letter, you indicate that it was the weight of the caterpillar (i.e. larval) stage that was being compared. However, you often refer to “S. littoralis size” in the text (L248, L252, L308, Table 1). Both of these terms are misleading as 1) S. littoralis (or “individuals” as found on L241) by itself could just as easily refer to all stages of development, and at no point in your methods or results (other than on the y-axis label of Figure 1b) do you make clear that you are exclusively comparing caterpillar weights, and 2) “size” is ambiguous and may consist of several different responses (e.g. body or wing length). Additional arguments can be made for issues with the “mortality” and “proportion pupated” as neither one of these are absolute values given the truncated experiment. Also, if for multiple plant species, no individuals had yet reached pupation at the time you ended the experiment, would development length for all of those caterpillars be the same despite the likelihood for differences to develop had the experiment continued? Ultimately, I indicated in my original report that a proper assessment of performance would likely require an additional assay (although not the “several concurrent attempts” suggested by the authors in their reply), and other than a significant revision to frame the performance responses appropriately, unfortunately the changes made are insufficient.
Additional Comments:
If I am interpreting your results correctly, your significance values in Table 2 are a function of comparing an empty model vs. a model parameterized with a single given predictor. Alternatively, Figure 1 depicts the actual relationships between predictor and response combinations deemed to be significant in your AIC analyses. However, there is a great deal of data scatter in some of the panels, and now that the statistics have been added, indeed some of these relationships appear to lose significance. Despite this, these relationships are treated as significant in your discussion. This inconsistency should be addressed.
If the proportion of individuals that a) made it to pupation, and b) suffered greater mortality is higher on the low quality plants, wouldn’t the makeup of your herbivore “populations”, and thus pressure and pattern of herbivory, vary by plant species? Accordingly, measurements of leaf damage aren’t simply a function of plant effects, but also changes in population structure of the feeding individuals. For many insect species, late instar larvae do substantial more damage than early instar individuals, so phenology certainly plays a role. The only way to have controlled for this would have been to release individuals of the same developmental stage onto plants for a short period and calculate damage accumulated in that window of time. This would be worth considering in the interpretation of your leaf damage results.
L364 Remove reference to probability of pupation as it was not statistically analyzed
L378 Figure 1a relates to spine toughness, not trichome density, and the relationship is negative, not positive as you indicate here
L387-388 In the "leaf damage" section below (L423), you claim that invertebrates have no issue avoiding spines, but here you indicate that there was a negative effect and make no attempt to explain it
L430-437 None of this speaks to the contrasting effects of spines on insect performance and leaf damage in the current study
Author Response
Dear reviewer,
thank you very much for your additional valuable comments that helped us improve our manuscript. We respond to all your comments. These all responses to individual questions are always given under each of your comments.
The authors have done a commendable job of improving several aspects of the manuscript. However, I remain unconvinced that the major issues have been adequately addressed. As mentioned, ending the experiment at the first sign of pupation severely hindered interpretation of the insect performance data as you do not have proper standardization for many of your response variables. For instance, you indicate that caterpillar development time varied by plant species. Accordingly, when you took fresh weight measurements, individuals from some plant species would be more advanced in their development than individuals from other plant species. In other words, you would end of with a phenology spectrum with different plant species containing different compositions of early-, mid-, and late-instar individuals. Because your measurement does not differentiate between these, the response conflates phenological variation in weight with stage-specific insect responses to plant quality. It would be much more appropriate to compare L6 weights, for example, or pupa weights, etc.
We think that this closely correlated with similar your comment below about the effect of later instar on leaf damage. So, we added this text to the Discussion: “The limiting factor for leaf damage in our experiment is the incomplete development of some caterpillars in the last instar. In this phenological stage, caterpillars usually cause the most damage, and therefore the effect of leaf damage can be biased.”
You are also not consistent with how you refer to your response variables. In your figure and your response letter, you indicate that it was the weight of the caterpillar (i.e. larval) stage that was being compared. However, you often refer to “S. littoralis size” in the text (L248, L252, L308, Table 1). Both of these terms are misleading as 1) S. littoralis (or “individuals” as found on L241) by itself could just as easily refer to all stages of development, and at no point in your methods or results (other than on the y-axis label of Figure 1b) do you make clear that you are exclusively comparing caterpillar weights, and 2) “size” is ambiguous and may consist of several different responses (e.g. body or wing length).
We agree that the term “S. littoralis size” was not the best, and since it always referred to the dry weight of the individual, we changed it in all places. We control all data, and we have the fresh weight of each individual before the experiment, and after that, we measured the fresh and also dry weight of it (more info in lines 168-170).
Additional arguments can be made for issues with the “mortality” and “proportion pupated” as neither one of these are absolute values given the truncated experiment. Also, if for multiple plant species, no individuals had yet reached pupation at the time you ended the experiment, would development length for all of those caterpillars be the same despite the likelihood for differences to develop had the experiment continued? Ultimately, I indicated in my original report that a proper assessment of performance would likely require an additional assay (although not the “several concurrent attempts” suggested by the authors in their reply), and other than a significant revision to frame the performance responses appropriately, unfortunately the changes made are insufficient.
We agree that this is a significant difference, but we think it did not fundamentally affect our results. If the reviewer would still see a significant difference in this, then this discrepancy could be resolved through survival analysis - analyzed as a day to pupation or death. That could solve everything correctly. If both the reviewer and the editor insist on it, we can add it.
Additional Comments:
If I am interpreting your results correctly, your significance values in Table 2 are a function of comparing an empty model vs. a model parameterized with a single given predictor. Alternatively, Figure 1 depicts the actual relationships between predictor and response combinations deemed to be significant in your AIC analyses. However, there is a great deal of data scatter in some of the panels, and now that the statistics have been added, indeed some of these relationships appear to lose significance. Despite this, these relationships are treated as significant in your discussion. This inconsistency should be addressed.
Single predictors are labeled by * in Table. The discrepancy may be that some factors have a p greater than 0.05 and do not have * in that table, but according to AIC, they improve the model. In the graphs, it fits with the table that is there from the beginning, but it was incorrectly marked as Table 3. However, if the reviewer insisted on a change, we can rephrase that without contributing, but it is not obvious.
If the proportion of individuals that a) made it to pupation, and b) suffered greater mortality is higher on the low quality plants, wouldn’t the makeup of your herbivore “populations”, and thus pressure and pattern of herbivory, vary by plant species? Accordingly, measurements of leaf damage aren’t simply a function of plant effects, but also changes in population structure of the feeding individuals. For many insect species, late instar larvae do substantial more damage than early instar individuals, so phenology certainly plays a role. The only way to have controlled for this would have been to release individuals of the same developmental stage onto plants for a short period and calculate damage accumulated in that window of time. This would be worth considering in the interpretation of your leaf damage results.
We added this text to the Discussion: “The limiting factor for leaf damage in our experiment is the incomplete development of some caterpillars in the last instar. In this phenological stage, caterpillars usually cause the most damage, and therefore the effect of leaf damage can be biased.”
L364 Remove reference to probability of pupation as it was not statistically analyzed
Removed.
L378 Figure 1a relates to spine toughness, not trichome density, and the relationship is negative, not positive as you indicate here
Sorry, a bad reference to Figure1a was given here, but it was supposed to be Table 2 (corrected it).
L387-388 In the "leaf damage" section below (L423), you claim that invertebrates have no issue avoiding spines, but here you indicate that there was a negative effect and make no attempt to explain it
In the sentence before, we specified that this protective element of trichomes applies to small individuals of similar size of aphids (lines 384-386).
L430-437 None of this speaks to the contrasting effects of spines on insect performance and leaf damage in the current study
In this section, we make no reference to the effect of spines on insect performance and leaf damage. Here we address the possible effect within the differences in the guilds of individual herbivores. I do not understand the reference to the mentioned paragraph.
Reviewer 3 Report
They have successfully addressed my comments.
Trichome literature can be improved even more (Lots of new papers this year), but that is just a minor comment.
Author Response
Dear reviewer,
thank you very much for your very kind comment.
They have successfully addressed my comments.
Trichome literature can be improved even more (Lots of new papers this year), but that is just a minor comment.
We fully agree that a large number of publications on this topic have been published this year, but I think that the number of cited articles is already very high.